# Investigation on the Interfacial Delamination of Glass Substrate Packaging Using Cohesive Zone Models

**DOI:** 10.3390/mi16080944

**Published:** 2025-08-18

**Authors:** Tianzuo Qin, Wen Yang, Qiqin Wei, Zhangsen Cen, Jianquan Chen, Yi Xie, Huiping Tang, Daoguo Yang

**Affiliations:** School of Mechanical and Electrical Engineering, Guilin University of Electronic Technology, Guilin 541004, China; tianzuo_qin@163.com (T.Q.); weiqiqin@guet.edu.cn (Q.W.); zhangsen_cen@163.com (Z.C.); jian_quanchen@163.com (J.C.); yi_xiexy@163.com (Y.X.); hui_ping_tang@163.com (H.T.); daoguo_yang@163.com (D.Y.)

**Keywords:** glass substrates, CZM, crack initiation and propagation, delamination assessment

## Abstract

This study aligns with the development trend of glass substrate packaging. The research aims to analyze the delamination of the substrate–adhesive layer-chip trilayer structure in packaging through experimental testing to obtain interface strength parameters. Subsequently, an iterative process combining experiments and simulations was applied to establish a cohesive zone model characterizing crack initiation and propagation. Finally, reliability analysis of the packaging structure was conducted. The results indicate that the load–displacement curves during sample loading can be experimentally acquired, enabling the determination of critical load values triggering interface delamination. The specific locations of delamination within the packaging structure are also clearly observed. Through simulation fitting, cohesive parameters reflecting interface strength are obtained, which serve as the basis for evaluating interface delamination fractures. Furthermore, applying the calibrated cohesive parameters to the established glass substrate model, simulation analysis evaluates delamination risks under thermal conditions.

## 1. Introduction

With the continuous advancement of modern industries, the electronic packaging industry has become a critical factor determining equipment performance and reliability. Although conventional packaging substrate materials such as FR4 and BT resin-based organic substrates have found extensive applications across various industrial scenarios, they still exhibit significant limitations in high-frequency, high-speed, high-density, and multifunctional integration packaging applications, including substantial signal transmission losses and inadequate coefficient of thermal expansion matching [1]. Glass substrates have emerged as a promising candidate for next-generation high-end packaging materials due to their excellent dielectric properties, thermal stability, and processing flexibility. These advantages demonstrate broad application potential in radio frequency devices (RF), micro-electromechanical systems (MEMSs), and optoelectronic packaging domains [2,3].

Despite the numerous advantages of glass substrate packaging, its practical implementation still faces several critical challenges. The inherent brittleness of glass materials renders them susceptible to mechanical shock and thermal cycling effects, leading to crack formation or even fractures during manufacturing, assembly, and service processes [4]. Furthermore, weak interfacial bonding characteristics between glass and heterogeneous materials persist as a fundamental limitation [5], which may induce reliability issues under thermal cycling or mechanical loading. An example in 2.5D/3D integrated packaging architectures, interfacial delamination between glass substrates and metallic fillers induced by coefficient of thermal expansion (CTE) mismatch can trigger micro-crack initiation and propagation at device interfaces, ultimately compromising the long-term reliability of the packaging system [6]. Furthermore, glass substrates face manufacturing process challenges, including technical bottlenecks in via formation and metallization techniques [7]. Consequently, achieving enhanced manufacturability and cost-effectiveness while preserving material performance remains a critical challenge for industrial-scale adoption.

Koji Fujimoto [8] proposed three types of copper-plated through-hole glass substrates, demonstrating superior thermal reliability and high-frequency performance through thermal simulation, thermal cycling tests, and RF characterization. Kaya Demir [9] employed a simulation–experimental hybrid approach to map the parametric sensitivity of failure modes, establishing a synergistic optimization framework between copper–glass interfacial stress distribution and mass production processes. Scott McCann [10] developed an integrated solution combining fracture mechanics-based finite element simulations with experimental validation, incorporating process-specific optimizations such as thin dielectric layer deposition, dicing process refinement, interfacial adhesion enhancement, and solder mask pullback compensation. Avinash A. Thakre [11] revealed the thickness-dependent energy release rate mechanism at gelatin hydrogel interfaces through direct shear testing, providing novel fracture mechanics insights into the adhesion-friction duality of soft solids. Tomas Serafinavicius [12] conducted comparative four-point bending creep tests on three interlayer materials polyvinyl butyral (PVB), ethylene–vinyl acetate (EVA), and soda–lime glass (SG) across 20–40 °C to evaluate long-term thermo-mechanical stability. Harsh Pandey [13] implemented localized surface texturing via electrochemical discharge machining (ECDM) and ultrasonic machining, systematically investigating surface roughness parameters on interfacial adhesion performance and establishing regulatory mechanisms for metallization reliability enhancement. Regarding the issue of IGBT bond wire interface delamination, Wu [14] et al. proposed a physics-based RUL prediction method utilizing chip temperature distribution modeling. They established a quantitative relationship between temperature fluctuations and crack propagation, constructing an equivalent electrical resistance model. Experimental results demonstrate prediction errors below 7%, with maintained robustness under competing solder layer degradation scenarios. Whereas these studies have advanced glass substrate processing and characterization techniques, the fundamental mechanisms governing interface delamination under operational thermo-mechanical stresses remain inadequately quantified.

In this paper, an approach combining experimental characterization with finite element simulation is proposed to conduct delamination analysis for the packaging interfaces. Investigation on the interfacial reliability challenges in glass substrate packaging is carried out. Through this methodology, a delamination risk assessment framework is established, thereby enabling quantitative evaluation of the failure mechanisms.

## 2. Experiments

The experimental study investigates interfacial fracture mechanics characteristics to explore the influence of external loading conditions on interfacial bonding strength. The experimental procedure consists of three stages: First, the single cantilever beam (SCB) testing method was adopted for Mode I interfacial strength evaluation. Geometrically consistent specimens were prepared accordingly; second, tensile tests were conducted to obtain load–displacement characteristic curves; third, we determined the critical fracture load and crack opening displacement at the delamination interface to calculate key parameters characterizing interfacial strength, such as critical energy release rate [15], using the following formula:(1)$$G_{C}=\frac{P_{C}^{2}}{2b}\frac{dC}{da}=\frac{P_{C}^{2}}{2b}\frac{4\lambda }{k}\left[\lambda^{3}a^{3}+{2\lambda }^{2}a^{2}+\lambda \right]$$(2)$$C=\frac{4\lambda }{k}\left[\frac{\lambda^{3}a^{3}}{3}+\lambda^{2}a^{2}+\frac{1}{2}\right];\;\lambda ={\left[\frac{3k}{E_{S}t_{s}^{3}b}\right]}^{\frac{1}{4}}$$(3)$$k=\frac{1-v_{f}}{\left(1-2v_{f}\right)\left(1+v_{f}\right)}\frac{E_{f}b}{t_{f}}$$

In the formula, $P_{C}$ denotes the critical load; the symbol a represents the initial crack length; b and t represent the substrate width and thickness, respectively; C and λ indicate beam compliance and wave number; k signifies the global stiffness modulus; E is the elastic modulus; and v represents Poisson’s ratio. Subscripts S and f in the equations correspond to the substrate layers and adhesive layers, respectively.

Experimental analyses revealed that bonding failures predominantly occurred within the adhesive layer, or at the adhesive–substrate interface. Given the inherent brittleness of silicon wafers and glass substrates in the test specimens, which are prone to fracture under external loading, the single cantilever beam test was selected as the experimental methodology based on material characteristics and specimen structural considerations. Recognized as an effective technique for evaluating Mode I interfacial fracture strength [16], the SCB test configuration (shown in Figure 1) comprises three primary components: the cantilever arm, adhesive layer, and substrate base. The experimental configuration primarily comprises three components: a cantilever beam, adhesive layer, and substrate base. An initial crack of length a is prefabricated at the interface. Through application of a load P at the free end of the cantilever beam, a relative displacement δ is generated under external loading conditions. This experimental approach enables quantitative determination of cohesive zone parameters governing interfacial crack propagation by monitoring delamination behavior under mechanical stress and analyzing load–displacement responses.

SCB testing was performed using a microcomputer-controlled electronic universal testing machine (Figure 2). The test was configured in displacement-controlled mode with a quasi-static tensile rate of 0.1 mm/min. During testing, the substrate side was clamped to the stationary grip, while the bonded silicon assembly was attached to the mobile loading grip. Concentricity between the specimen and loading axis was verified via laser alignment prior to testing to ensure data acquisition accuracy.

Figure 3 presents the tensile test data obtained from SCB experiments. Analysis of the experimental data reveals consistent load–displacement curves across specimens during the crack initiation phase, whereas significant variability emerges in the crack propagation stage. Experimental observations attribute this divergence to rapid interfacial separation upon reaching the critical load, resulting in post-failure curve dispersion. As this study focuses on characterizing interfacial delamination behavior, primary emphasis is placed on the critical load parameters at incipient separation, with post-failure states excluded from analysis. The extremum points of load–displacement curves are statistically summarized in Table 1. The mean critical load is 19.677 N with a corresponding displacement of 254.126 μm. The experimental data collected establishes a robust empirical foundation for the analytical evaluation of interfacial bonding strength.

Upon obtaining the maximum load parameters of each specimen, the Mode I critical energy release rate at the interface was calculated using the analytical formulations (1)–(3) derived from SCB testing. The required input parameters for these calculations are detailed in Table 2, with the resulting critical fracture energy summarized in Table 3. The magnitude of the energy release rate serves as a quantitative indicator of a material’s resistance to crack propagation, enabling preliminary determination of the critical energy threshold for delamination initiation at substrate interfaces [17]. When the energy release rate at the crack tip attains this critical threshold, interfacial delamination or crack propagation will occur. Figure 4 presents the Weibull distribution analysis of specimens at a 95% confidence level. The shape parameter (β) of 15.97 indicates strong consistency in energy release rate data, with data points tightly clustered around the scale parameter. The characteristic life parameter (η) of 246 slightly exceeds the median and mean values, reflecting mild positive skewness in the distribution. This signifies a 63.2% probability of failure when the energy release rate reaches 246 J/m^2^. Overall, the experimental data demonstrates excellent reliability characteristics.

## 3. Simulation of SCB Test

In interfacial delamination studies, the primary objective is to acquire interface strength parameters, necessitating the determination of the critical stress value that induces crack initiation. Experimental methods are used to directly capture dynamically evolving interfacial stress distributions due to continuous variations in the crack propagation zone area during delamination. Consequently, numerical simulations are required to establish the relationship between interfacial stress and separation displacement. Specifically, precise interface strength values must be obtained through a combined experimental–simulation approach, where parameter inversion is implemented by aligning experimental and simulated load–displacement curves [18]. Additionally, computational determination of critical energy release rates enables validation of simulation methodology and model applicability through comparative analysis with experimentally derived values.

The study established an interfacial delamination simulation model within the static structural module of ANSYS Workbench 2022 platform (Figure 5), maintaining geometric consistency with the SCB test specimen configuration. Based on the experimental model, an aluminum alloy tensile fixture is bonded using AB glue with a controlled thickness of 0.1 mm. Critically, the AB glue’s bonding strength must exceed that of UV adhesives to ensure experimental delamination occurs specifically at the adhesive–substrate interface. The computational model comprises five material components: monocrystalline silicon wafer, glass substrate, epoxy adhesive layer, aluminum alloy tensile fixture, and two-component epoxy bonding layer. Detailed material properties for each constituent are specified in Table 4.

Given the simulation’s focus on interfacial failure behavior, a contact–separation algorithm was implemented to model delamination processes. Cohesive contact pairs were inserted at pre-defined critical interfaces, with contact constraints enforced via the penalty method. The interface damage evolution was governed by critical separation distance and maximum normal stress parameters, determined through iterative optimization of simulation–experimental correlation. During Mode I cohesive zone model parameterization, pure Mode I debonding conditions were enforced for interface separation simulation. To ensure consistency between simulation and SCB experimental data, maximum normal contact stress was iteratively optimized through simulation–experimental fitting, while strictly controlling the contact gap at complete interface separation to match experimentally measured critical displacement. In alignment with Mode I loading characteristics, Mode II parameters were set to zero in the constitutive model.

In the simulation boundary condition setup, one end of the aluminum alloy fixture is fixed while a displacement of 0.3 mm is applied to the opposite end. The experimental tensile rate is 0.1 mm/min, and the end time of the simulation step is set to 180 s through conversion. As mesh density is a critical parameter in finite element analysis, rational mesh settings play a decisive role in the accuracy of simulation results and computational efficiency. The study employs a 3D mesh model utilizing hexahedral second-order elements with a global size of 0.3 mm. Specifically, the adhesive layer retains a vertical dimension of 0.01 mm, discretized into three layers vertically. The chip and substrate are partitioned into four and five vertical layers, respectively. This differentiated mesh configuration ensures precision in interfacial stress analysis while effectively controlling computational resource consumption, achieving an optimal balance between simulation efficiency and accuracy.

Following mesh configuration and parameter initialization, the study implemented iterative optimization of model core parameters through a simulation–experimental fitting methodology. Coordinated adjustment of maximum normal contact stress and interface stiffness achieved convergence between experimental and simulated load–displacement curves, as illustrated in Figure 6. After 6–10 iterative calculations, the simulated critical normal failure stress demonstrated agreement with experimental test values, validating the effectiveness of the parameter optimization methodology.

The same calibration methodology was applied to the remaining experimental specimen curves, with fitting results shown in Figure 7. The slope of the simulated curve in the figure exhibits a slight deviation from the experimental curve slope. This occurs because parameter values such as stiffness are iteratively adjusted during fitting to align the simulated peak load and displacement with experimental values, and stiffness modifications alter the initial slope of the fitted curve. Cabello, M. [19] attributes slope deviations to damage-induced reduction in apparent stiffness within flexible adhesives. In this article, Cabello, M. [20] proposes that stiffness varies with deflections, where deflection adjustments modify stiffness and affect curve superposition; experimental curves exhibit slight fluctuations and nonlinearity during crack initiation, potentially attributable to assembly clearance between specimens and fixtures or misinterpretation of sensor fluctuations as material nonlinearity. Simulations inherently avoid such artifacts, maintaining clear linearity in simulated curves; at peak load, experimental curves enter the crack propagation phase where load-bearing capacity rapidly declines until complete fracture, whereas simulation curves exhibit smoother transitions, due to simulation software employing idealized treatment of instability processes, and require numerical algorithm stability. However, as this research focuses on interfacial strength parameters, propagation-stage deviations remain within acceptable tolerance thresholds. Table 5 presents extremum parameters of the load–displacement curves. Analysis reveals a simulated average peak load of 19.6888 N, consistent with experimental values, while the corresponding average critical displacement is 254.352 μm.

By analyzing the load–displacement curves during crack initiation, the maximum normal contact stress and separation distance causing interfacial cracks can be determined. The simulation results are substituted into the bilinear cohesive zone model for calculation, yielding Mode I interfacial strength parameters with specific values listed in Table 6. The deviation between the simulated mean critical energy release rate and the experimentally calculated mean value from Table 2 is 0.277 J/m^2^, demonstrating the validity of the simulation method and model applicability.

Through the combined methodology of experimental testing and numerical simulation modeling, the critical interfacial stress and critical energy release rate at the chip-adhesive–substrate critical interface under Mode I loading conditions were determined. The corresponding mean values yield the cohesive zone model parameters listed in Table 7. This set of CZM parameters characterizes the interfacial strength of the chip-adhesive–substrate heterogeneous interface at the T0 stage.

## 4. Simulation on a Package Structure

In practical applications, encapsulated devices undergo prolonged exposure to complex operating conditions including high humidity and thermal cycling, where harsh environmental factors can induce reliability issues such as interfacial delamination. This chapter conducts a preliminary investigation using 2D finite element simulations to preliminary analyze glass substrate packaging structures, leveraging the calibrated cohesive parameters characterizing interfacial strength obtained previously. The research focuses specifically on delamination phenomena within practical glass substrate packaging architectures under service environments. The objective is to apply the established interface constitutive relationships to engineering-relevant packaging models, enabling systematic assessment and prediction of interfacial delamination risks and failure mechanisms under thermal loading.

Compared to 3D models, 2D models significantly simplify mesh generation, substantially enhance computational efficiency, and offer simpler modeling procedures. The 2D finite element method (FEM) is particularly suitable for investigating plane stress conditions and conducting parametric sensitivity analyses, and is thus selected as the modeling approach for preliminary reliability analysis of glass substrates in this study. Figure 8 illustrates the 2D simulation model configuration, with geometric features and material properties summarized in Table 8. The simulation implements contact debonding constraints through calibrated cohesive parameters applied as virtual layers at critical glass-adhesive interfaces. Given the adhesive’s high sensitivity to thermal variations, delamination behavior was analyzed under temperature cycling from 120 °C to 25 °C.

In the simulation post-processing, the study uses damage values to assess delamination at critical interfaces. In Workbench, the contact debonding damage evolution parameter is an advanced parameter in the damage-based contact debonding model, controlling the rate of the exponential damage evolution law. It defines the damage accumulation rate from initiation to complete interfacial failure, with a damage value of 1 indicating full failure. Figure 9 displays interfacial delamination analysis from the prior chapter’s tensile simulation using the damage debonding model. Post-processing results show a damage variable of 0.86 at edge regions of critical interfaces, confirming entry into the failure phase with delamination crack initiation.

Figure 10a presents simulation results of the specific model under temperature variations. After inserting the cohesive parameter module at critical interfaces, the equivalent stress becomes minimal at these surfaces while reaching maximum values at the upper surface. This phenomenon likely occurs because the separation tendency caused by thermal expansion at interfaces with contact debonding modules is no longer forcibly constrained, allowing contact pressure to decrease or even vanish. When constraining forces at contact interfaces significantly reduce or disappear, the system must redistribute strain energy from thermal expansion through alternative mechanisms, consequently increasing equivalent stress on the upper surface—consistent with physical principles and FEM contact mechanics. Simultaneously, Figure 10b reveals that the contact debonding damage evolution parameter shows a persistent damage value of 0 at critical interfaces, indicating no delamination occurs during temperature cycling.

## 5. Conclusions

An integrated experimental and simulation approach is employed to investigate the delamination behavior in glass substrate–adhesive-chip trilayer structures, and the following conclusions are drawn:

(1) Tensile testing yields load–displacement curves during specimen loading, enabling determination of the interfacial delamination threshold load at 19.677 N, with failure predominantly occurring within the adhesive or at the adhesive–substrate interface.

(2) Calibration of cohesive parameters characterizing interfacial strength through simulation–experimental fitting yields a maximum normal contact stress of 3.374 MPa and critical energy release rate of 238.0718 J/m^2^, demonstrating model and simulation methodology validity.

(3) Preliminary simulations of the package structure suggest that after inserting the cohesive parameter module at critical interfaces, the equivalent stress becomes minimal at these surfaces while reaching maximum values at the upper surface—consistent with physical principles and FEM contact mechanics, and there was no interfacial delamination at critical interfaces under thermal variations when analyzed using the contact debonding damage evolution parameter.

## Figures and Tables

**Figure 1 micromachines-16-00944-f001:**
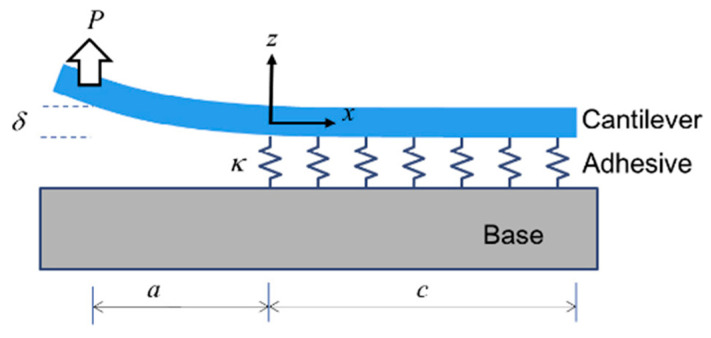
Single cantilever beam experiment [16], where “a” denotes the prefabricated crack length and “c” indicates the adhesive bond length between substrate and adhesive. Reprinted with permission from Ref. [16]. Copyright 2015, Elsevier Ltd.

**Figure 2 micromachines-16-00944-f002:**
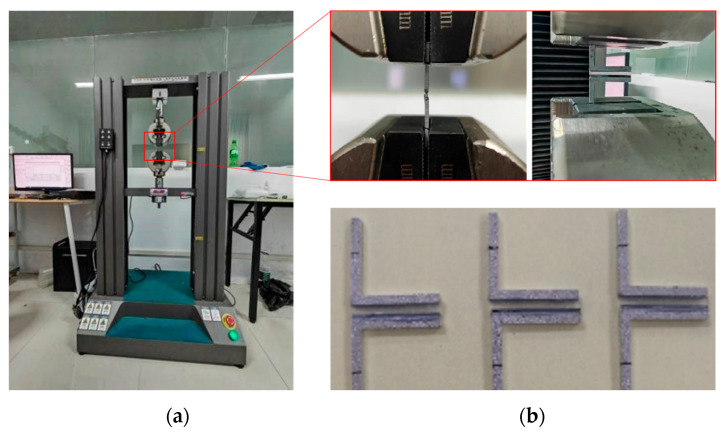
(**a**) The test setup; (**b**) test samples.

**Figure 3 micromachines-16-00944-f003:**
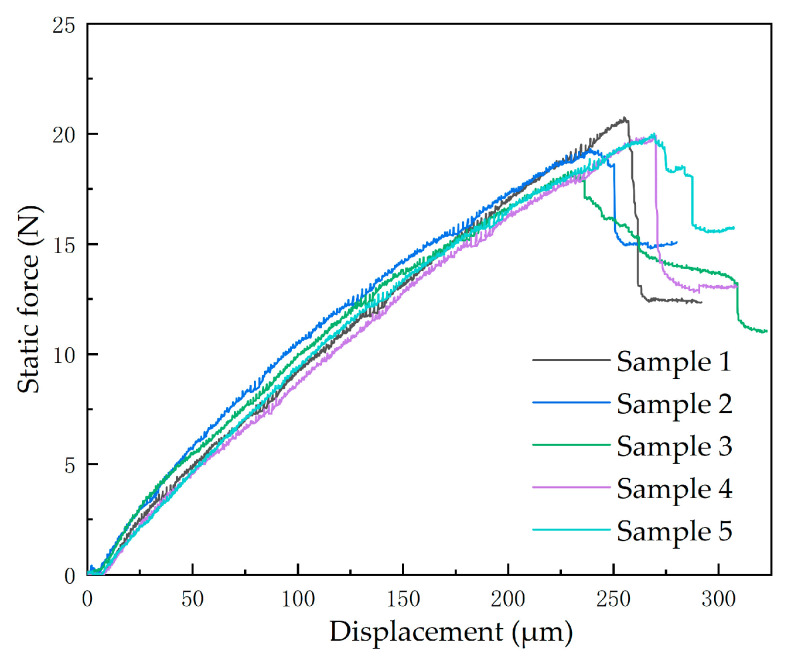
The load–displacement curves.

**Figure 4 micromachines-16-00944-f004:**
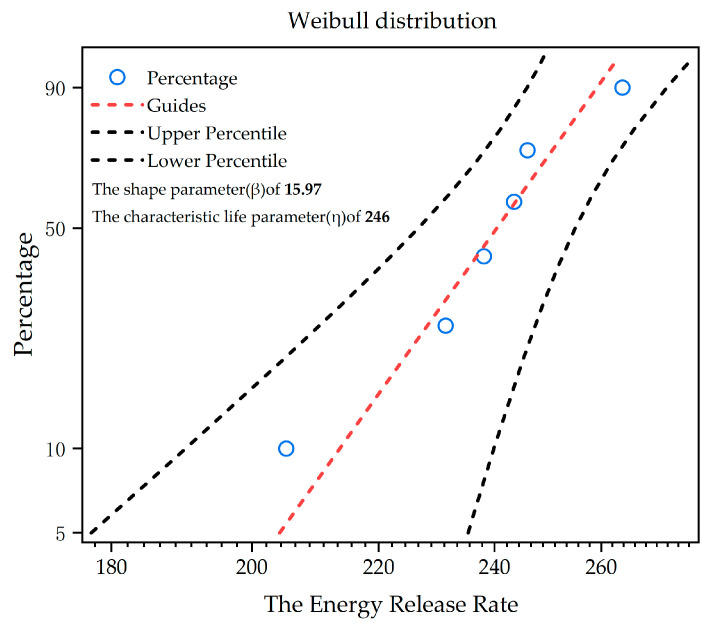
Schematic diagram of Weibull distribution.

**Figure 5 micromachines-16-00944-f005:**
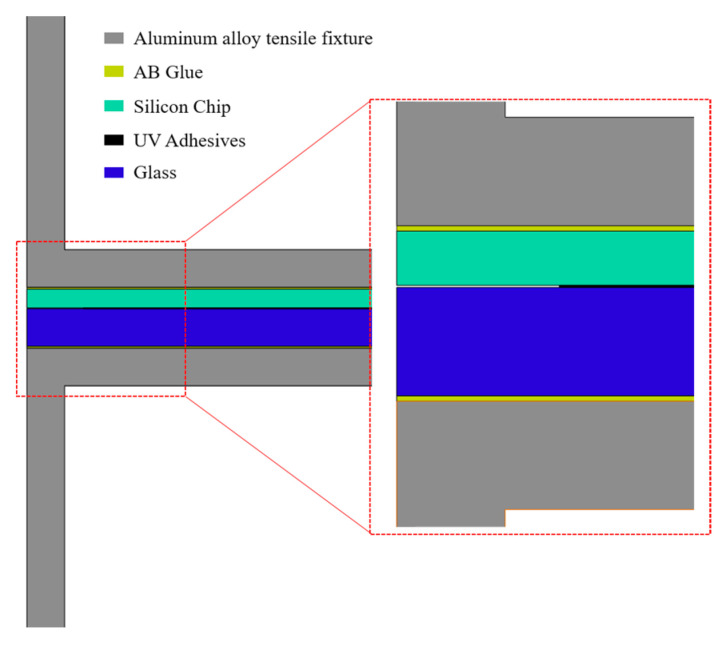
Simulation stretching model.

**Figure 6 micromachines-16-00944-f006:**
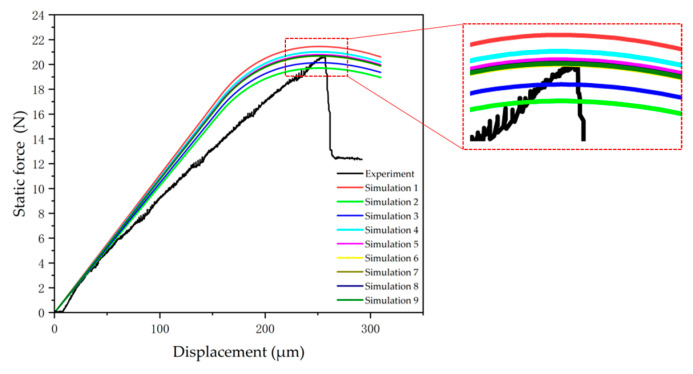
Simulation fitting process.

**Figure 7 micromachines-16-00944-f007:**
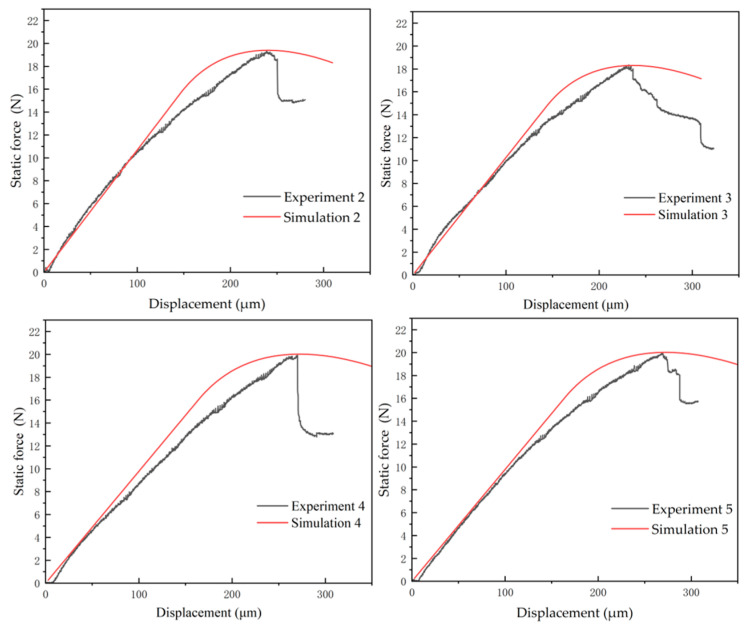
Simulated fitting curve.

**Figure 8 micromachines-16-00944-f008:**
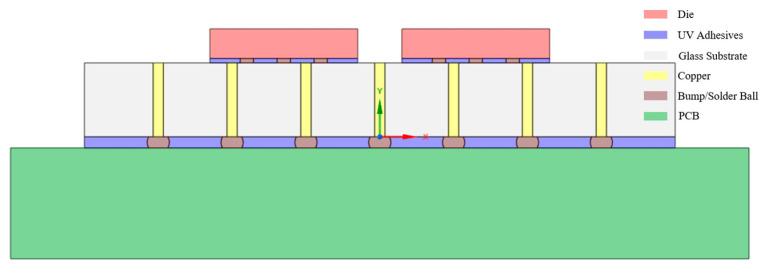
Two-dimensional model modeling.

**Figure 9 micromachines-16-00944-f009:**
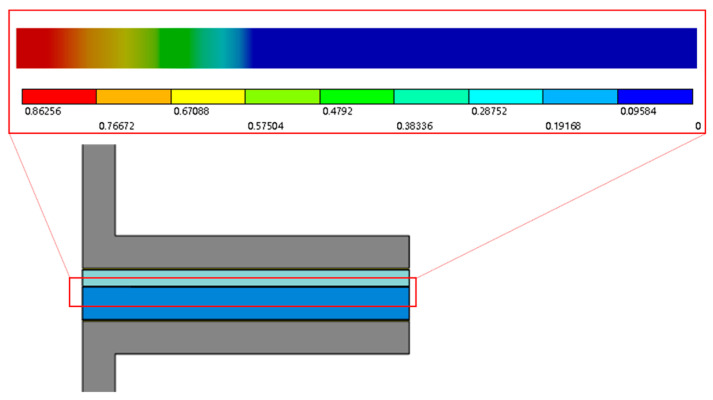
Stratification of the interfacial stratification of the stretched sample.

**Figure 10 micromachines-16-00944-f010:**
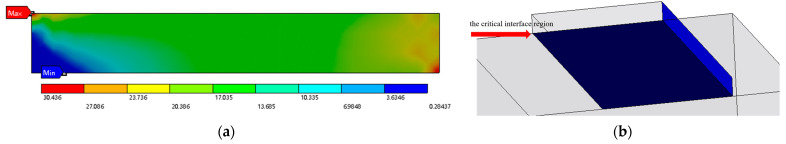
(**a**) Isostressing force contour diagram of adhesives layer interface; (**b**) delamination result diagram of substrate and adhesive layer interface.

**Table 1 micromachines-16-00944-t001:** Load–displacement curve extremum.

Number	Maximum Load (N)	Displacement (µm)	Load/Displacement (N/µm)
Sample 1	20.746	256.14	0.081
Sample 2	19.414	239.78	0.081
Sample 3	18.286	235.2	0.078
Sample 4	19.919	270.13	0.074
Sample 5	20.020	269.38	0.075
Average	19.677	254.126	0.078

**Table 2 micromachines-16-00944-t002:** Critical energy release rate calculation parameters.

$P_{C}$ (N)	a (mm)	b (mm)	$t_{s}$ (mm)	$t_{f}$ (mm)	$v_{s}$	$v_{f}$	$E_{s}$ (GPa)	$E_{f}$
Table 1	3	1	1	0.04	0.27	0.3	162.7	7.6

**Table 3 micromachines-16-00944-t003:** Critical energy release rate.

Number	Critical Fracture Energy (J/m^2^)
Sample 1	264.189
Sample 2	231.353
Sample 3	205.250
Sample 4	243.546
Sample 5	246.021
Average	238.072

**Table 4 micromachines-16-00944-t004:** Material parameters.

Parts	Geometric Dimensions	Young’s Modulus (MPa)	Poisson’s Ratio	CTE (1/°C)
	Length × Width × Height (mm)			
Si	20 × 1 × 1	162,700	0.27	2.578 × 10^−6^
Glass Substrates	20 × 1 × 2	69,930	0.2149	9.35 × 10^−6^
AB Glue	20 × 1 × 0.1	1800	0.3	4 × 10^−5^
UV Adhesives	17 × 1 × 0.04	7600	0.3	2.65 × 10^−5^

**Table 5 micromachines-16-00944-t005:** Simulated load–displacement extremes.

Number	Experiment		Emulation	
	Maximum Load (N)	Displacement (µm)	Maximum Load (N)	Displacement (µm)
Sample 1	20.746	256.14	20.701	252
Sample 2	19.414	239.78	19.401	239.1
Sample 3	18.286	235.2	18.302	235.48
Sample 4	19.919	270.13	20.02	272.59
Sample 5	20.020	269.38	20.02	272.59
Average	19.677	254.126	19.6888	254.352

**Table 6 micromachines-16-00944-t006:** Simulated critical energy release rate.

Number	Maximum Normal Contact Stress (MPa)	Contact Gap (µm)	Critical Fracture Energy (J/m^2^)	Experiment-Simulation Differences (J/m^2^)
Sample 1	3.628	252	263.044	1.145
Sample 2	3.367	239.1	231.044	0.310
Sample 3	3.135	235.48	205.610	0.359
Sample 4	3.37	272.59	246.022	2.476
Sample 5	3.37	272.59	246.022	0.001
Average	3.374	254.352	238.348	0.277

**Table 7 micromachines-16-00944-t007:** Sample cohesion parameter.

Maximum Normal Contact Stress (MPa)	Contact Gap (µm)	Critical Fracture Energy (J/m^2^)	Artificial Damping Coefficient
3.374	254.352	238.3483	1 × 10^−8^

**Table 8 micromachines-16-00944-t008:** Material geometry and parameters.

	Geometric Dimensions	Young’s Modulus (MPa)	Poisson’s Ratio	CTE (1/°C)	Tg (°C)
	Length × Height (mm)				
Die	2 × 0.4	162,700	0.27	2.578 × 10^−6^	
Glass Substrates	8 × 1	69,930	0.2149	9.35 × 10^−6^	
Bump/Solder Ball Adhesives	2 × 0.06/ 8 × 0.15	7600	0.3	2.65 × 10^−5^/ 5.032 × 10^−5^	90
PCB	10 × 1.5	24,600	0.136	1.55 × 10^−5^	
Copper	0.14 × 1	126,000	0.345	1.674 × 10^−5^	
Bump	0.18 × 0.06	75,800	0.35	2.4 × 10^−5^	
Solder Ball	0.26 × r = 0.3	58,900	0.4	2.4 × 10^−5^	

## Data Availability

Data are contained within the article.

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
