# Peer review of "Investigation on the Interfacial Delamination of Glass Substrate Packaging Using Cohesive Zone Models"

_micromachines, 2025, doi:10.3390/mi16080944_

Round 1
Reviewer 1 Report
Comments and Suggestions for Authors
The paper describes characterization and modeling of Si-glass adhesive bonded pair. The experimental test is using a single cantilever beam method. Hybrid modeling with FEA is carried out on the testing to establish a modeling approach, which is then applied for a preliminary simulation of a package structure.
The paper is well written and the topic is relevant for the audience. There are some suggestions to make a good paper even better:
- Page 2: “Although numerous research on the glass substrate has been done…”: It is advised to rephrase this sentence expressing the problem statement specific to this paper.
- Page 2, last sentence: it is advised to divide the sentence, and first introduce the testing method (SCB), and the specimen, then the equation for the critical energy release rate.
- Related to the formula of G_c, the important cantilever arm parameter ‘a’ was not defined in the text. It is also advised to provide a reference e.g. [15] to the formulas for the SCB test.
- In Fig. 1, parameter ‘b’ should be the width, not the length parameter. In the caption, it could be: “single cantilever beam experiment”.
- Page 3: “Interfacial delamination analyses…” Experimental analyses revealed, that bonding failures predominantly occurred within the adhesive layer, or at the adhesive-substrate interface.
- Figure 2: pictures are too small, and a detailed picture of the real bonded assembly is missing, which would be important for the schematics in Fig. 4.
- Table 2: in the table header, the notation should be the same as introduced at the formula.
- Section 3, title may be improved to e.g. “Simulation of SCB test”.
- Page 6: simulation descriptions:
- “simulation time of 180 s”: it should be noted, that this is a virtual time, as this model is time independent.
- “hexahedral second-order element mesh”: it suggest, that a 3-D mesh was used, this should be stated explicitly.
- Figure 4: “stretch the bracket” should be rephrased; and the gluing to the top and bottom bracket should be assessed in the text.
- Figure 5-6: a discussion is needed on the following:
- the initial slope of the simulated curves deviates vs. the experimental ones,
- the characteristics of the simulated curves are different, the bending of the curves near the maximum needs explanation,
- the slight non-linearity of the measured curves needs explanation.
- Table 6: the calculated average value of the difference is not right considering the values in the last column.
- Page 8: last sentence: “temperature variations”: it should be noted that the adhesive is highly sensitive for temperatures.
- Figure 7: figure caption should be rephrased.
- Figure 8: figure caption should be rephrased. Furthermore, this figure belongs to the previous section.
- Conclusions seem unfinished. In conclusion (3), it should be emphasized, that the statement belongs to a preliminary simulation of a package structure.
Reviewer 2 Report
Comments and Suggestions for Authors
This study addresses reliability challenges in next-generation glass substrate packaging by investigating interfacial delamination of substrate-adhesive-chip trilayers. Through experimental single cantilever beam (SCB) testing,interfacial strength parameters are quantified by analyzing load-displacement curves to identify critical delamination loads and failure locations. An iterative experiment-simulation calibration process establishes cohesive zone model (CZM) parameters that accurately characterize crack initiation and propagation. Generally, I agree that the paper nicely fits in the scope of Micromachines and can be accepted for publication after some mandatory revisions.
1 Please further analyze the discrepancy between the simulated and measured results in fig. 6
2 Please comment on the 2-D simplification and its accuracy, why dont use 3-d modeling
3 It seems that all the material properties lack temperature dependence
4 might need to Perform Weibull analysis of critical fracture energy (Table 3).
5 should implement fatigue damage accumulation model for thermal cycling, also please clearly indicate the condition.
6 reference should be added and more papers should be cited. recent advance in Interfacial Delamination study in electronic packaging should be cited and discussed, for example
Wu, "Novel Prognostics for IGBTs Using Wire-Bond Contact Degradation Model Considering On-Chip Temperature Distribution," in IEEE Transactions on Power Electronics, vol. 40, no. 3, pp. 4411-4424, March 2025.
7 no comparisons with other model parametrization method, which might undermine the novelty of your proposal here.
8 Please add more reliability analysis results, as characterize crack initiation and propagation is to predict its failure. An example calculation would be great.
Round 2
Reviewer 2 Report
Comments and Suggestions for Authors
no further comments